# Prepared to Accompany the End of Life during Pandemics in Nursing Homes [note 1]

**DOI:** 10.3390/jcm11206075

**Published:** 2022-10-14

**Authors:** Norbert Krumm, Cordula Gebel, Lars Kloppenburg, Roman Rolke, Ulrich Wedding

**Affiliations:** 1Department of Palliative Medicine, University Hospital Aachen, RWTH Aachen University, 52074 Aachen, Germany; 2Department of Internal Medicine II, Division of Palliative Medicine, University Hospital Jena, 07747 Jena, Germany

**Keywords:** palliative care, nursing home, isolation, emotional support, pandemic

## Abstract

Background: The COVID-19 pandemic confronted nursing homes with a variety of challenges to ensure the provision of palliative care for residents. *PallPan-Implement* aimed to adapt the recommendations of the National Strategy for the Care of Seriously Ill, Dying Adults and their Families in Times of Pandemic (PALLPAN) in such a way that nursing facilities can use and implement them. Methods: Based on 33 PALLPAN recommendations, we developed a questionnaire, conducted a pilot implementation for selected nursing homes, and asked for qualitative feedback. Results: The developed questionnaire contains 22 main questions. A three-stage pilot implementation with an introductory event, processing phase, and evaluation event took place in seven facilities. The facilities evaluated the developed questionnaire as helpful. Feedback from the facilities identified three major categories: (a) requirements for facilities should be realistic to avoid frustration, (b) the creation of a pandemic plan for palliative care only is impractical, (c) measures for the psychosocial support of staff is particularly necessary, but was perceived as difficult to implement. Conclusions: The practical implementation of recommendations requires a concept and material tailored to facilities and areas. The strategy of PallPan Implement developed in this project appears to be target-oriented, well-received, and can be recommended for further implementation.

## 1. Introduction

During the COVID-19 pandemic, nursing homes were particularly challenged in many ways to maintain standards of care. Advanced age and comorbidities are the major risk factors for a severe course of the disease after a SARS-CoV-2 infection [1]. Therefore, nursing homes care for a highly vulnerable group of people. Reports of outbreak events in such facilities and the associated number of deaths have sadly made this clear [2,3].

Against this backdrop, there were extensive bans on visits to nursing homes [4], as well as a strong reduction in contact within these facilities, both between staff and between staff and residents, as well as between residents themselves and residents and visitors, in the course of the pandemic. Continuing contact was realized in strict compliance with hygiene regulations, i.e., with the use of personal protective equipment for staff and visitors. Access to the facilities was also limited for external caregivers and volunteers, which additionally led to a partial deficit in care, e.g., regarding specialist medical care or social activities [5].

The observance and implementation of guidelines on contact restrictions and hygiene instructions in this area were accompanied by a particularly increased workload [6]. On the one hand, pandemic-related protective measures had to be consistently adhered to in order to protect residents who were particularly vulnerable to the viral infection. On the other hand, these measures had to be implemented under the conditions of a tense and increasingly worsening staff situation as well as in premises not designed for quarantine measures. The results of overwork and other burdens among staff working in nursing homes are well documented [5,7]. In addition, there were frequent updates with only comprehensive information on new regulations. Review and interpretation in relation to the individual facility, and ultimately their implementation, also represented a new workload. Due to their frequency and sometimes contradictory nature, such regulatory updates contributed to a feeling of uncertainty (“What may currently be done and how?”) among referring health care professionals [8].

Notably, there was a media presence of reported cases of death in loneliness or isolation, not only but also in nursing homes, cases that presumably led to great suffering for those affected, but also for the staff, some of whom had been caring for the deceased residents for a long time [9]. At an early stage of the pandemic, some authorities recommended to transfer patients with COVID infection from nursing homes to hospitals, to protect other residents, irrespective of the medical need, the patients’ will, and the already existing overload of work in hospitals.

To address such unmet needs for patients and proxies, health care professionals and regulatory authorities, the PallPan consortium was founded with support from the German Network University Medicine (NUM) that was funded from the German Ministry of Education & Science (BMBF) in 2020. The PallPan consortium invented a nationwide strategy for delivering palliative care in times of pandemic. A core part of this strategy is a list of 33 palliative care recommendations based upon 16 studies for the care of seriously ill and dying people and their relatives in Germany—consented to by means of a Delphi procedure [10,11].

In view of the effects and experiences described above, it seemed reasonable and necessary to refine the PallPan recommendations specifically with regard to their concrete applicability for facilities dealing with nursing home patients. A team of researchers from the University Hospitals of Aachen and Jena has dedicated itself to this project within the framework of the PallPan follow-up project for the concrete implementation of the PallPan recommendations in nursing homes.

## 2. Aim

The overall goal of this implementation project is to promote the applicability and implementation of the more or less general PallPan recommendations to the special situation of nursing homes and thus to ensure good palliative care for nursing home residents and their relatives, even in times of pandemic. Nursing home residents and staff members should be able to benefit from the knowledge and experience gained during the COVID-19 pandemic, which was used to develop the PallPan recommendations, without being presented with concrete adaptation processes that may not be suitable for their individual facilities. The aim is to support nursing home staff members to develop individual solutions within their nursing home to maintain their good palliative care structures in times of pandemic, based on the findings of the PallPan studies.

## 3. Methods

The present implementation project was conducted during the so-called third wave of the COVID-19 pandemic in Germany, from September to December 2021. To realize the transfer of theoretical knowledge into practice, in a first phase we developed a tool, “PallPan recommendations for nursing homes”, by tailoring the PallPan recommendations to nursing homes’ situations. In a second phase, we tested the material created for this purpose in a “pilot implementation phase” in a number of selected nursing homes in two cities.

### 3.1. Phase 1: Development of the Tool “PallPan Recommendations for Nursing Homes”

#### 3.1.1. Background

The adaptation processes in nursing homes regarding the preparation of their palliative care structures for pandemics is complex. Special consideration had to be placed on the fact that facilities for the care of elder persons have different prerequisites regarding size, equipment, and the structural set-up of palliative care or also the regional integration.

Furthermore, recommendations are usually brought to organizations from the outside. The resulting need to adapt certain processes is presented to those involved [12]. Specific questions were derived from the PallPan recommendations that are based on systemic-constructivist theories aiming to stimulate adaptation processes from within the elder care facilities and to build a bridge between theory and practice in the specific nursing home [13,14]. The tool questions were designed to enable an active discussion of the recommendations within this selected nursing home. Based on these core questions, these facilities had to start an analysis of the internal structures to ultimately perform a description of adaptation strategies.

#### 3.1.2. Procedure

The research team first decided to identify, out of the 33 PallPan recommendations [11], those that have a relevance for institutions of care for the elderly (see Appendix A). For this purpose, the recommendations were independently judged in terms of their relevance for and references to the care of the elder persons by three researchers (CG, LK and NK). The creation and development of the questionnaire was iterative and step-by-step. All participants of the research group derived questions based on the recommendation. The concrete applicability and unambiguity of the questions were of importance. A first version was compiled and discussed in the author group (UW, RR, CG, LK and NK). The preliminary version of the questionnaire was then discussed with an external expert (Graduate pedagogue with many years of experience in the implementation of hospice and palliative care in nursing homes). Feedback was incorporated until the final product was ready for pilot testing. Based on their important comments, the questionnaire was adapted and expanded. A new section on strengthening the institutions’ self-efficacy was included (Table 1, Section B).

### 3.2. Phase 2: Pilot Implementation Phase

#### 3.2.1. Procedure

The piloting of the tool was carried out in a three-stage process. At a kick-off event (1), the background (PallPan recommendations) and the aim of the project were explained, and the tool was presented. The tool was then handed over to the individual facilities, which worked on it in the subsequent phase of filling out the form (2). For this purpose, the formation of a “processing committee” of staff involved in palliative care was suggested. During this phase, the research team was always available to the facilities for any questions or comments that might arise. After 2–6 weeks (in relation to the date of the kick-off event and depending on the completion time predicted by the facilities), a final feedback event (3) took place, where on the one hand the question of which adaptations were established in the care facilities was reflected on and on the other hand a critical reflection on the tool took place. The further implementation of their specific recommendations was not part of the pilot implementation phase. Both the kick-off event and the final feedback event are designed as a workshop. If a workshop was not possible due to the pandemic situation, the events were conducted digitally or by telephone. Two member of the research group conducted (NK and LK) all events.

#### 3.2.2. Participants

The recruitment of specific institutions to participate in the project was carried out by means of convenience sampling. Institutions were approached personally during network meetings and asked about their willingness to contribute to the implementation phase. Participants from the individual institutions were management staff, coordinators, nursing service management staff and employees of the care facilities. In Germany, different general practitioners (GPs) care for nursing home residents. No physician was involved in the pilot testing. The participation was voluntary.

#### 3.2.3. Data Collection and Analysis

After each event, qualitative feedback was collected from the participants, in which the facilities could give their feedback on (a) the applicability and usefulness of the tool and (b) the general structure and formulation of the tool. The results of the exchange and discussion during the individual events were incorporated directly afterwards as a field protocol in previously developed protocol templates. The team (NK, LG, CG) then categorized the data material via a thematic analysis and discussed the results.

## 4. Results

### 4.1. Phase 1: Development of the Questionnaire Tool

Twenty-five of the thirty-three PallPan recommendations were identified as being of particular importance for nursing homes and were integrated as questions in the tool “PallPan recommendations for nursing homes” (see Appendix A). We discussed the questionnaire further and revised it within the research team. Finally, the questionnaire comprised 20 main questions, allocated into four main topics.

(1)Best possible palliative care for residents in times of pandemic(2)Integration of relatives in times of pandemic(3)Protecting employees in times of pandemic(4)Supporting employees

The structure of the tool is presented in Table 1. The complete tool can be provided by the authors.

Supplementary and thematically appropriate sub-questions were assigned to each of these themes. Subsequently, we decided to integrate further sections into the tool.

### 4.2. Phase 2: Pilot Implementation

Seven institutions (characteristics are shown in Table 2) with a total of 18 participants (characteristics are shown in Table 3) participated in the implementation events. In total, 14 events took place, six on site, two in digital format, and six by telephone. During the processing phase, none of the facilities expressed any questions, comments, or concerns about support needs vis-à-vis the research team. In the context of the final event, it was emphasized on the one hand that the facilities saw their palliative care structures confirmed to a large extent by the tool, but on the other hand ideas for adapting their own palliative care structures and the questionnaire were also mentioned. The facilities participating in the pilot project generally rated the tool as easy to use, helpful and goal-oriented.

In the following, we summarize some examples of feedback after working with the tool:

The tool was perceived as self-explanatory.

*“The goal is already clear from the title”* (fieldnotes LK)

People who completed the questionnaire felt supported in their reflections on the conditions in their facilities.

*“Questions gave self-reflection support”* (fieldnotes NK)

There was skepticism about integrating learned lessons into existing pandemic plans, and participants wondered if another location would be more convenient.

*“Integration of findings into palliative care concept (no [extra] pandemic palliative care plan)”.* (fieldnotes LK)

In addition, the fact that questions had a confirming effect was praised.

*“A lot of things were already going quite well and that was nice to notice”* (fieldnotes LK)

Feedback was extensively discussed and analyzed in the research group (CG, LK, NK), and three central themes occurred:Requirements for facilities should be kept realistic (also under the status quo of the provision of resources), since otherwise stressful experiences occur when filling out the questionnaire.The development of a palliative care pandemic plan within facilities was not considered practicable. Findings should rather be integrated into existing documents/concepts, which are also consulted.Measures for the psychosocial support of staff should be strengthened within the facility, and the facilities would be grateful for further concrete approaches/guidance on pandemic-friendly and resource-saving implementations.

## 5. Discussion

This research project developed a model for implementing palliative care recommendations in nursing homes during pandemics. Part of the strategy was to adapt the more general PALLPAN recommendation to a more nursing home-specific instrument, to include staff members for the adaption of the nursing home-specific instruments, and to make recommendations that fit the situation of the specific nursing home. This helps the nursing home avoid another pandemic situation in which they feel unprepared and receive public criticism for not delivering good palliative care in a better way. The next steps after this pilot should be the involvement of more nursing homes—nursing homes with less palliative care experience. However, everybody hopes that the final proof of the effectiveness of the strategy—better palliative care in nursing homes in a pandemic situation—will never be tested. The tool has the structure of a catalogue of questions, preceded by an introductory text and ending with a selection of topic-related best-practice examples to inspire concrete adaptation processes. The main block of questions on “Assessment of palliative care in the respective facility in times of pandemic”, consisting of 20 main questions, is preceded by two further blocks of questions on “Experiences during the COVID-19 pandemic” and on “Structure of the facility’s internal palliative care”. This structure helped in staying focused during the implementation process in the respective nursing homes.

Various studies highlighted the complexity of implementation processes in nursing homes, in and outside a pandemic situation [15,16]. Barriers and facilitators have also been described in the implementation of palliative care approaches in geriatric care [17,18]. Due to the resources and time constraints of the COVID-19 pandemic and the simultaneous need for adaptation processes in nursing homes, we developed a parsimonious model for the implementation of recommendations for action. This is because both resource scarcity and time scarcity affect not only the care provided in elder care facilities but also the adaptation process that is now needed (Nilsen 2018). To effectively implement End-of-Life programs in nursing homes, the inclusion of the context of the facility, the respective professionals, and an appreciation of the culture of the home related to palliative care is essential [17].

Participants reaffirmed that measures for the psychosocial support of staff should be strengthened within the facilities. Billings [19]) found that frontline health and social care workers are likely to need a flexible system of support including peer, organizational and professional support.

Facilities would be grateful for further concrete approaches/guidance on pandemic-friendly and resource-saving implementations. Many questions and recommendations are related and complementary, so that a palliative culture, provided and maintained outside of the pandemic, is a good basis for caring adequately for residents during times of pandemic. A vision for palliative care shared by all staff would be helpful [18].

## 6. Limitations

The aim of this implementation was an adaption and first pilot of the implementation of the PALLPAN recommendations for specific nursing homes. We are aware that this pilot implementation could be planned and conducted on a deeper scientific basis. Against the background of a high need, after the experience of the first and second waves, and during the beginning phase of the third wave, the implementation should start as soon as possible to support nursing homes in this situation.

Regarding limitations of the present study, the authors are aware of a selection bias. Primarily, only committed facilities were involved in this research. They were addressed and asked to participate within existing networks of collaboration between palliative care and geriatric nursing care. This could result in overly positive feedback on the project. In addition, strategies to maintain good palliative care in times of a pandemic are based on their existence prior to the pandemic. Furthermore, the authors would like to highlight that all work on the study was accomplished within a very short period of four months (September–December 2021) due to the ongoing pandemic situation and existing needs, as well as financing issues from the grant provider. PallPan Implement shall finally contribute to a better maintenance of palliative care in pandemic situations in nursing homes. This pilot test did not analyze the extent of palliative care that nursing home residents actually received.

## 7. Conclusions

In our view, PallPan Implement is suitable for strengthening the resilience of individual employees and entire teams, enabling dignified palliative care at the end of life in the pandemic for those affected and for their families and loved ones.

The concrete implementation of normative recommendations via the development of a catalogue of questions based on these recommendations appears to be goal-oriented and could contribute to an increased acceptance and individualized implementation of the recommendations for palliative care under pandemic conditions. The present concept enabled individual thematic work by means of processing the question catalogue even under the conditions of scarce resources. After incorporating the feedback from the facilities participating in the pilot, the developed tool will be made available to other nursing homes in Germany, e.g., via the PallPan website (https://pallpan.de (accessed on 10 June 2022)). For further accessibility, the institutions participating in the pilot made various suggestions, including publicity in nursing journals as well as via actors involved in palliative care (hospices, clinics, churches, pastoral care, etc.) and via actors involved in political decisions (city, state, health authorities, etc.). In the future, other areas of palliative care (SAPV, AAPV, hospice services, etc.) should also, in a similar way, be enabled to concretely apply the recommendations of the PallPan joint project in their area of activity and their concrete individual form of organization. Furthermore, the procedure of a practical implementation of theoretical findings based on a catalogue of questions also seems to be applicable to other research areas.

## Figures and Tables

**Table 1 jcm-11-06075-t001:** Structure of the tool for palliative care in times of pandemic.

Section		Aims and *Examples*
A General introduction of the background and the aim of the tool		The inclusion of an “introductory text”, which explains the aims and background of the tool, was intended to make the tool easier to understand and use.
B Questionnaire		
	General questions on dealing with the COVID-19 pandemic	To establish a link to the COVID-19 pandemic and to demonstrate the relevance of the tool, an introductory block of questions on internal experiences during the COVID-19 pandemic was also included. *Example: What has worked in your facility to deal with the pandemic in terms of dealing with the seriously ill and dying, their families, and your own staff?*
	Reflection questions on the internal structure of palliative care within the facility	It intends to facilitate a comparison of the findings on how good palliative care can be organized during a pandemic in the individual facility with the actual existing conditions. *Example: In general, how do you identify residents with palliative care needs in your facility?*
	Questions derived from the PallPan recommendations for action	Twenty main questions to facilitate the implementation of the different PallPan recommendations for action. The questions were allocated into 4 topics: 1. Best possible palliative care for residents in times of pandemic *Example: With which palliative care actors is it imperative to maintain cooperation even in times of pandemic in order to provide residents with the best possible palliative care?* 2. Integration of relatives in times of pandemic *Example: How would you maintain pandemic-compliant communication with relatives?* 3. Protecting employees in times of pandemic *Example: What personal protective equipment (PPE) would you stock to protect the employees?* 4. Supporting employees *Example: What measures for grief counseling for your employees and other psychosocial support are feasible in times of a pandemic?*
C Best practice examples		A selection of best practice examples published within the framework of the PallPan recommendations was assigned to the individual questions to inspire possible adaptation processes. *Example: Conduct supervision/coaching in compliance with hygiene regulations (e.g., small group, outdoor, digital, hybrid).*

**Table 2 jcm-11-06075-t002:** Characteristics of participating care facility (*n* = 7).

Size	*n*
<80 residents	3
>80 residents	4
COVID breakouts	
Yes	5
No	2
Location	
Thüringen	4
North Rhine Westphalia	3

**Table 3 jcm-11-06075-t003:** Characteristics of participants.

Number of Participating Persons per Facility	Round Table 1 (*n*)	Round Table 2 (*n*)
A	2	2
B	3	3
C	5	2
D	5	3
E	1	1
F	1	1
G	1	1
Function		
Facility Manager	3	
Palliative Care Nurse	1	
Hygiene Officer	1	
Ward manager	8	
Care manager	4	
Social worker	1	

## Data Availability

Exclude this statement as the study does not provide any publicly archived datasets are available.

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
