# Peer review of "Prepared to Accompany the End of Life during Pandemics in Nursing Homesâ€"

_jcm, 2022, doi:10.3390/jcm11206075_

Round 1
Reviewer 1 Report
the paper needs a better decription in discussion and conclusion about effective strategies deriving from the implementation of the tool.
It's not clear (but it could depend on the fact that I don't know how exactly are managed nursing homes in Germany, since I'm not German) whether Physicians were involved and how.
I think could be useful to attach as a table the tool
Reviewer 2 Report
Thank you very much for the opportunity to review this manuscript which I found interesting and compelling. I have very liitle in the way of criticism. The Table 1 should be reformatted to make it easier to read and understand.
On line 105 I think you have omitted the word 'persons' after elder...
On line 139 I am unsure exactly what the filling phase is - perhaps this could be phrased in a different way to ensure it is understood?
Reviewer 3 Report
Thank you for the opportunity to review this manuscript. It is very difficult to comment as I need much more background to the PALLPAN recommendations; the evaluations process and analysis methodology. There did not appear to be any attachments or appendices for me to refer to and it appears the manuscript where I may find details of the PALLPAN recommendations is either in press or in German. As the questionnaire developed was based on these recommendations I cannot comment on the validity of the project. Examples of what further details is needed is as follows: Information on the participating facilities. What were their experiences of the PALLPAN recommendations - where they implemented in their facilities; when; for how long; What was the level of knowledge of the PALLPAN recommendations that enabled them to respond to the questionnaire. What method of implementation science was used to assess the recommendations - is there a specific model? Who were the members of the expert group? Who was the external expert - what was their qualifications? What feedback was received and incorporated - were there many changes? Is there a copy of the questionnaire itself? More information is required on the 3 stage process of piloting e.g. what was the final feedback event - a workshop? How was the qualitative feedback collected - interviews? What form of analysis was undertaken - what theory was used? What was the ad hoc model? I regret that in its present form there is insufficient details to comment on the findings.
